

# Egg clutch dehydration induces early hatching in red-eyed treefrogs, *Agalychnis callidryas*

María José Salica[1], James R. Vonesh[2,*] and Karen M. Warkentin[3,4,*]

[1] CONICET, Instituto de Ecorregiones Andinas (INECOA), Universidad Nacional de Jujuy, San Salvador de Jujuy, Argentina
[2] Department of Biology, Virginia Commonwealth University, Richmond, VA, United States of America
[3] Department of Biology, Boston University, Boston, MA, United States of America
[4] Smithsonian Tropical Research Institute, Panamá, República de Panamá
[*] These authors contributed equally to this work.

## ABSTRACT

Terrestrial eggs have evolved repeatedly in tropical anurans exposing embryos to the new threat of dehydration. Red-eyed treefrogs, *Agalychnis callidryas,* lay eggs on plants over water. Maternally provided water allows shaded eggs in humid sites to develop to hatching without rainfall, but unshaded eggs and those in less humid sites can die from dehydration. Hatching responses of amphibian eggs to dry conditions are known from two lineages with independent origins of terrestrial eggs. Here, we experimentally tested for dehydration-induced early hatching in another lineage (*Agalychnis callidryas,* Phyllomedusidae), representing a third independent origin of terrestrial eggs. We also investigated how dehydration affected egg and clutch structure, and egg mortality. We collected clutches from a pond in Gamboa, Panama, and randomly allocated them to wet or dry treatments at age 1 day. Embryos hatched earlier from dry clutches than from wet clutches, accelerating hatching by ∼11%. Clutch thickness and egg diameter were affected by dehydration, diverging between treatments over time. Meanwhile, mortality in dry clutches was six-fold higher than in control clutches. With this study, early hatching responses to escape mortality from egg dehydration are now known from three anuran lineages with independent origins of terrestrial eggs, suggesting they may be widespread. Further studies are needed to understand how terrestrial amphibian eggs can respond to, or will be affected by, rapid changes in climate over the next decades.

## INTRODUCTION

Terrestrial eggs have evolved repeatedly in many species of teleost fishes and amphibians (*Martin & Carter, 2013*). In tropical anurans, *Gomez-Mestre, Pyron & Wiens (2012)* found 48 independent origins of terrestrial reproduction. The evolution of terrestrial breeding may be driven by the risk of aquatic predation in early life stages (*Duellman & Trueb, 1986*; *Touchon, 2012*). However, nonaquatic reproduction also entails risks. Terrestrial eggs are exposed to different threats than those affecting aquatic eggs, including terrestrial predators (*Warkentin, 1995*; *Warkentin, 2000*), pathogens (*Warkentin, Currie & Rehner, 2001*), and

Corresponding author
María José Salica,
mjsalica@gmail.com

the novel threat of dehydration (*Mitchell, 2002*; *Touchon & Warkentin, 2009*). The risk of egg dehydration most strongly affects species without parental care, and this threat could be exacerbated by climate change (*Donnelly & Crump, 1998*). As well as temperature, rainfall patterns are changing in the tropics. Specifically, even if overall rainfall remains similar, in the Neotropics rainfall events are becoming less frequent, resulting in an increase in dry spells during the rainy season (*Hulme & Viner, 1998*; *Christensen et al., 2007*; *Allan & Soden, 2008*). Therefore, it is important to understand the potential responses of vulnerable life stages to such climate variations.

Environmentally cued variation in hatching time is widespread in many taxa (*Warkentin, 2011a*) and serves as an important defense mechanism against egg-stage risks. Environmentally cued hatching (ECH) is well documented in anurans (*Warkentin, 2011b*); much of this research addresses biotic threats to eggs and larvae, and a substantial subset addresses responses of embryos to hypoxia. The terrestrial eggs of red-eyed treefrogs, *Agalychnis callidryas,* one of the most studied species, hatch early in response to multiple environmental threats, including predator attack (snakes, *Warkentin, 1995*; wasps, *Warkentin, 2000*), fungal infection (*Warkentin, Currie & Rehner, 2001*) and flooding, which can kill embryos too young to hatch (*Warkentin, 2002*). Embryos presumably use some of the same mechanisms to respond to these different risks. For instance, all responses require a means to exit from the egg and the ability to regulate expression of this process (*Cohen, Seid & Warkentin, 2016*). Nonetheless, different types of threat provide very different types of cues. Their detection requires different sensors, and assessing different risks may require different cognitive mechanisms. Thus, embryos that respond to one threat, using one type of cue, may be insensitive to other cues and unresponsive to other threats.

Only a few studies of ECH have examined how amphibian eggs respond to drying conditions (*Warkentin, 2011b*); thus, it is unclear how widespread hatching responses to egg dehydration might be. To date, such responses are known from two lineages with independent origins of terrestrial eggs deposited on vegetation above water in rainforest environments. In the treefrog *Dendropsophus ebraccatus* (Hylidae: Dendropsophinae), eggs exposed to dehydration hatch earlier and more synchronously than well-hydrated clutches (*Touchon & Warkentin, 2010*; *Touchon, Urbina & Warkentin, 2011*). In the glassfrog *Hyalinobatrachium fleischmanni* (Centrolenidae: Hyalinobatrachinae), fathers hydrate their developing embryos during dry weather. When the caring parent is removed, increasing risk of egg dehydration, the embryos also respond by hatching earlier and more synchronously (*Delia, Ramírez-Bautista & Summers, 2014*). Here, we tested for dehydration-induced early hatching in another lineage (*Agalychnis callidryas*, Phyllomedusidae), representing a third independent origin of terrestrial eggs. We also investigated how dehydration affected egg and clutch structure, and egg mortality.

## MATERIALS & METHODS

### Study system

The recently redescribed family Phyllomedusidae (Amphibia: Anura: Arboranae, *Duellman, Marion & Hedges, 2016*) are uniformly terrestrial egg layers. They place eggs on vegetation

over water, into which tadpoles fall upon hatching. These treefrogs have evolved several strategies to avoid egg dehydration. Females absorb water from their environment before oviposition and deposit eggs surrounded by well hydrated jelly (*Pyburn, 1970*; *Pyburn, 1980*). In addition, some species wrap eggs in a funnel-shaped nest of leaves, and surround their eggs with eggless jelly capsules as water reservoirs (*Faivovich et al., 2010*). Nonetheless, after the eggs are deposited, embryos must face dehydration and other risks with no further parental assistance. *Agalychnis callidryas* inhabits lowland wet forest from the Yucatan through Panama (*Frost, 2016*), breeding in seasonal ponds and swamps. This species lays their gelatinous egg masses exposed on vegetation, without wrapping them in leaves. Maternally provided water allows shaded eggs in humid sites to develop to hatching without rainfall. However, unshaded eggs and those in less humid sites can die from dehydration. We studied them at the Smithsonian Tropical Research Institute in Gamboa, Panama. At this locality egg mortality from dehydration has historically been low but detectable (e.g., 3% in 1998, vs. zero at a pond in Corcovado Park, Costa Rica, in 1993 and 1994; *Warkentin, 2000*; *Gomez-Mestre & Warkentin, 2007*). However, in the extremely dry El Niño of 2015 many entire egg cohorts laid in Gamboa perished from dehydration (K Warkentin, pers. obs., 2015).

## Experimental design

We collected 30 healthy egg clutches (38.8 ± 3.6 eggs (mean ± SE)) laid on the night of 24 July 2011 from the Experimental Pond in Gamboa, Panama (9°07′15′N, 79°42′14′W). All clutches were collected with the leaves on which they were laid, mounted on plastic cards for support and attached to the sides of plastic cups in a vertical orientation. Each cup contained aged tap water to catch hatched tadpoles. Eggs in each clutch were counted, and any dead or undeveloped eggs (possibly unfertilized) were noted. Clutches were randomly allocated to a wet treatment or a dry treatment starting at age 1 day. Wet clutches were heavily sprayed with aged tap water multiple times daily, taking care not to overspray onto dry clutches. Dry clutches were unsprayed or minimally sprayed in some cases where eggs were dying from dehydration. Clutches were located on the same table in a laboratory with a mean temperature of 26.8 °C (range: 25.5–28.5 °C), and mean humidity of 82.4% (range: 78–88%); nearby ponds under rainforest canopy cover are usually slightly cooler and more humid. Clutches were maintained on a 12:12 light: dark photoperiod, based on local sunrise/sunset times. All clutches were checked for hatching at least five times daily. Clutches were photographed daily with a ruler for egg size measurements, from age 1 to 4 days. At each age, for each clutch, we measured two orthogonal diameters for each of 10 eggs from the photographs, using ImageJ (NIH); for analysis, we used the average of the two diameters. We also measured the thickness of each clutch when it entered the experiment at 1 day old and after two days in the treatments, at 3 days old, by inserting a fine probe orthogonally through the thickest part of the clutch, between eggs, to the leaf surface. This measurement included both eggs and associated jelly thickness.

## Statistical analysis

Analyses were conducted using generalized linear models followed by likelihood ratio or *F* tests implemented using R v. 3.3.1 (2016-06-21; *R Development Core Team, 2011*). We used

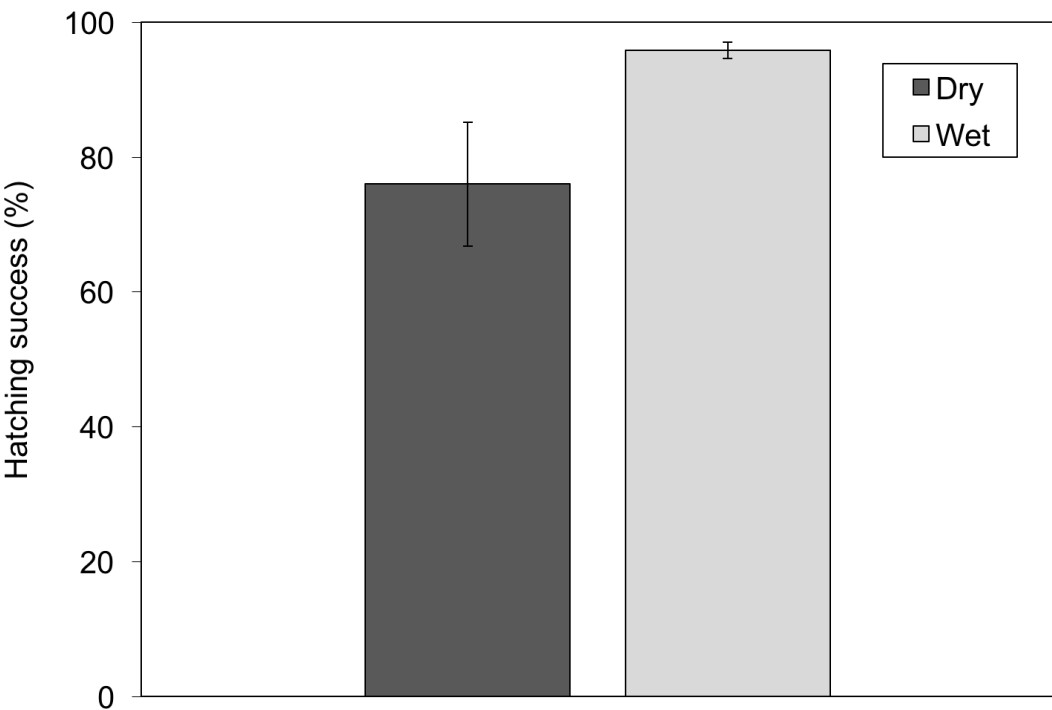

**Figure 1** **Hatching success of well-hydrated *Agalychnis callidryas* embryos was significantly higher than that of drying embryos.** Data are mean percent of embryos that hatched successfully (±SE across 15 clutches per treatment).

logistic regression with binomial errors to test whether the hydration treatment altered the proportion of embryos that survived to hatching. As we focused on embryonic mortality from desiccation, this analysis excluded embryos that showed no initial development (e.g., presumed unfertilized). We used linear models with normally distributed errors to test for the independent and interactive effects of the hydration treatment and clutch size on time to hatching (h) and of hydration treatment and days post-oviposition (dpo) on egg diameter (mm) and clutch thickness (mm).

## RESULTS

Mean survival was significantly lower in clutches from the dry treatment compared to the wet treatment ($X^2 = 6.86$, $df = 1, 28$, $P = 0.009$, dispersion parameter $= 15.2$; Fig. 1). Embryonic mortality averaged $24.0 \pm 0.9\%$ (mean $\pm$ SE, here and throughout) in the dry treatment compared to only $4.0 \pm 0.1\%$ in the wet treatment. Mortality in the dry treatment was also more variable, ranging from zero to 100%. Desiccation mortality occurred early in development; embryos that achieved hatching competence prior to desiccation were able to hatch and escape further drying.

We assessed timing of hatching at three time points along the hatching curve; (1) initiation of hatching, (2) half of the clutch hatched, and (3) completed hatching. Initiation of hatching depended only on hydration treatment (hydration: $F_{1,24} = 9.76$, $P < 0.01$, clutch size: $F_{1,24} = 2.04$, $P = 0.16$, hydration $\times$ clutch size: $F_{1,24} = 0.08$, $P = 0.8$). Embryos
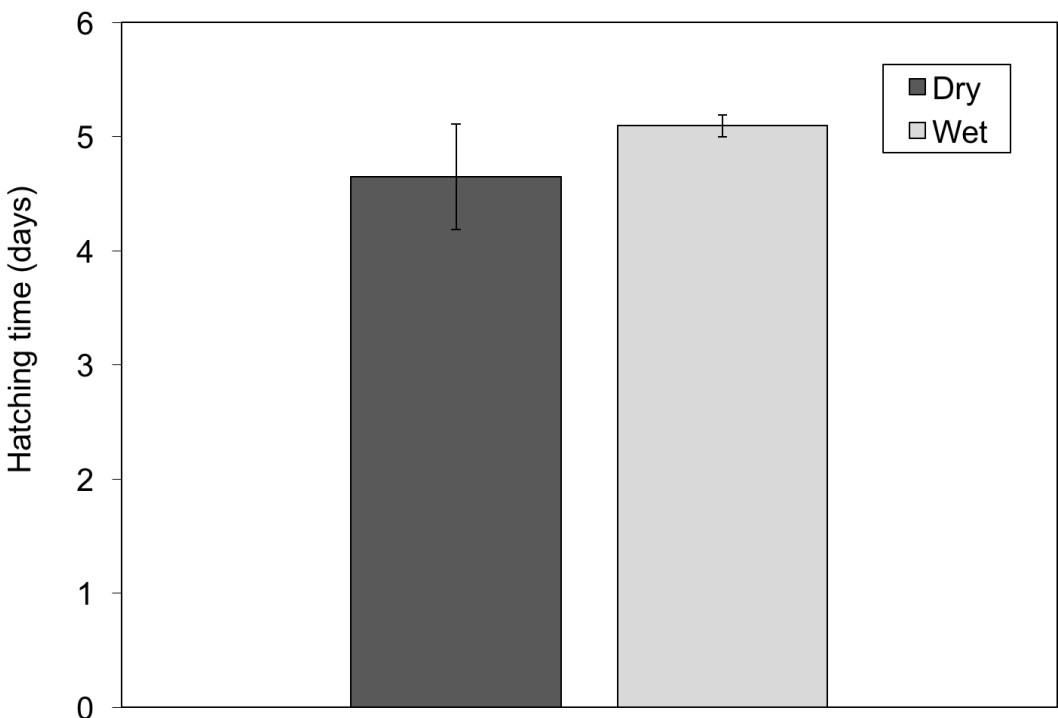

**Figure 2** *Agalychnis callidryas* **embryos in dry clutches hatched earlier than did those in wet clutches.** Data shown are the mean age when clutches started hatching, ±SE across 15 clutches per treatment.

from the dry treatments started hatching $10.73 \pm 3.4$ h earlier than the wet treatment (Fig. 2). Time for half of the embryos to hatch depended on both hydration and clutch size (hydration: $F_{1,24} = 14.1$, $P < 0.001$, clutch size: $F_{1,24} = 6.05$, $P = 0.02$, hydration $\times$ clutch size: $F_{1,24} = 0.08$, $P = 0.78$). Wet clutches reached the 50% hatch point $16.3 \pm 10.3$ h later than dry clutches and each additional egg in a clutch increased time to half hatch by $0.29 \pm 0.14$ h. Time to hatch completely was similarly dependent on hydration and clutch size (hydration: $F_{1,24} = 12.7$, $P = 0.0015$, clutch size $F_{1,24} = 5.64$, $P = 0.026$, hydration $\times$ clutch size interaction: $F_{1,26} = 0.34$, $P = 0.57$). Wet clutches finished hatching $19.9 \pm 11.1$ h later than dry clutches and each additional egg increased time to complete hatching by $0.33 \pm 0.15$ h. In both treatments, hatching was gradual and asynchronous, but the entire hatching curve was earlier in the dry treatment (Fig. 3).

Mean egg diameter was a function of the interaction between hydration and days post-oviposition (hydration: $X^2 = 33.93$, $df = 1$, $P < 0.001$; dpo: $X^2 = 12.26$, $df = 1$, $P < 0.0004$; hydration $\times$ dpo: $X^2 = 15.91$, $df = 1$, $P < 0.001$, Fig. 4). Initially, in both dry and wet clutches, egg diameters increased due to absorption of water from the egg jelly into the perivitelline space; however, wet eggs swelled more rapidly. Eggs in wet clutches continued to swell, then stabilized in diameter at age 3 days. By contrast, from 2 days eggs in dry clutches shrank, with the difference between treatments increasing over time.

Mean clutch thickness also was a function of the interaction between hydration and days post-oviposition (hydration: $X^2 = 3.12$, $df = 1$, $P = 0.077$; dpo: $X^2 = 0.48$, $df = 1$,

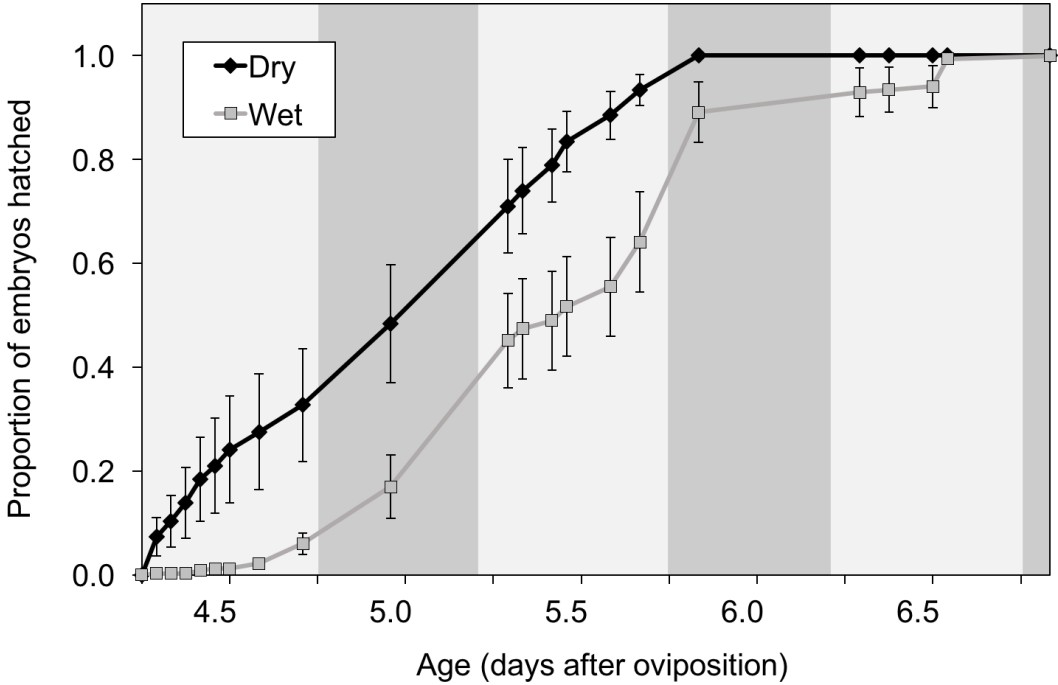

**Figure 3** *Agalychnis callidryas* **embryos hatched ≈11% earlier from drying vs. wet clutches.** Data are mean proportion hatched at each age (±SE across 15 clutches per treatment), of all that eventually hatched. Dark and light shading along the *x*-axis indicates photoperiod.

$P = 0.49$; hydration $\times$ dpo: $X^2 = 6.05$, $df = 1$, $P = 0.014$, Fig. 5). At the beginning of the experiment, at age 1 day, there was no difference in thickness between clutches assigned to different treatments (dry: $7.03 \pm 1.70$ mm, wet: $6.73 \pm 1.53$ mm). However, two days later wet clutches were much thicker than dry clutches (dry: $5.67 \pm 1.63$ mm, wet: $7.50 \pm 1.84$ mm).

## DISCUSSION

Our results show that red-eyed treefrogs can accelerate hatching when exposed to the gradual threat of dehydration over embryonic development. In this study, the acceleration in hatching timing (11%) was less than that reported for other frogs (*Dendrosophus ebraccatus*: 17%, *Touchon & Warkentin, 2010*; *Hyalinobatrachium fleishmanni*: 59%, *Delia, Ramírez-Bautista & Summers, 2014*). It may be that, compared with those species, *A. callidryas* has a relatively limited capacity to accelerate hatching under the threat of drying. Indeed, based on field monitoring of eggs, both *D. ebraccatus* and *H. fleishmanni* both appear at higher risk of mortality from dehydration than does *A. callidryas*. Dehydration led to 98% mortality in terrestrial eggs of *D. ebraccatus* exposed to lack of rainfall during the first 48 h post-oviposition (*Touchon & Warkentin, 2009*). Similarly, in male removal experiments generating "orphan" clutches of *H. fleishmanni*, 78% of total mortality was due to dehydration (*Delia, Ramírez-Bautista & Summers, 2013*). Alternatively, because the mortality imposed by our drying treatment was moderate (24%), compared with the possible risk of mortality under more extreme weather conditions, it may not have tested the limits of *A. callidryas* capacity to accelerate hatching.

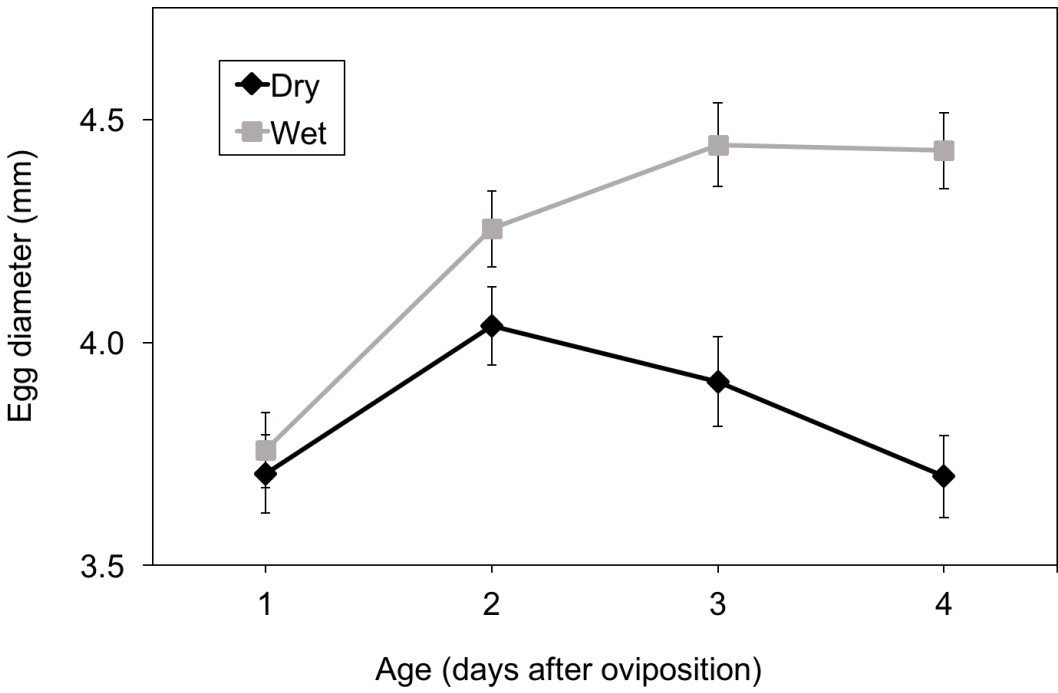

**Figure 4** **Effect of development and hydration treatment on *Agalychnis callidryas* egg diameter.** Data are mean ± SE across 15 clutches per treatment. Egg diameter was a function of the interaction between hydration treatment and age.

The hatching pattern of drying clutches—accelerated but gradual hatching, over a period of days—was very similar to the hatching pattern of clutches infected by a pathogenic fungus which caused about 40% mortality and 17% acceleration of hatching (*Warkentin, Currie & Rehner, 2001*; *Warkentin, 2011b*). Both fungus and dehydration are chronic threats that affect egg clutches gradually and potentially provide cues over extended periods of development. However, what those cues are, or how embryos detect them, is in both cases unknown. Red-eyed treefrog embryos use physical disturbance or vibrations to assess danger in predator attacks (*Warkentin, 2005*) and respond by hatching very rapidly, within seconds (*Cohen, Seid & Warkentin, 2016*; *Warkentin et al., 2007*). They also use hypoxia as a cue to hatch from eggs that are flooded, responding to submergence in minutes (*Warkentin, 2002*). Like fungus infection, dehydration does not move eggs, and neither threat has a sudden, acute onset. Either vibrational cues or another sudden change in clutch conditions may be necessary to induce rapid or synchronous hatching.

Both clutch thickness and egg diameter were affected by dehydration, diverging between treatments over time. Dehydration began to affect these variables from age 3 days, when both clutch thickness and egg diameter decreased in dry treatment eggs. Our results suggest that during early developmental stages water moves from the jelly layers into the perivitelline space, enlarging the eggs (*Salthe, 1965*), as diameter of the vitelline chamber increased even in the dry treatment. Later in embryonic development (from three days), after available water from jelly layers has been absorbed, the eggs can absorb additional water from external sources, such as rainfall. Without external sources of water, egg

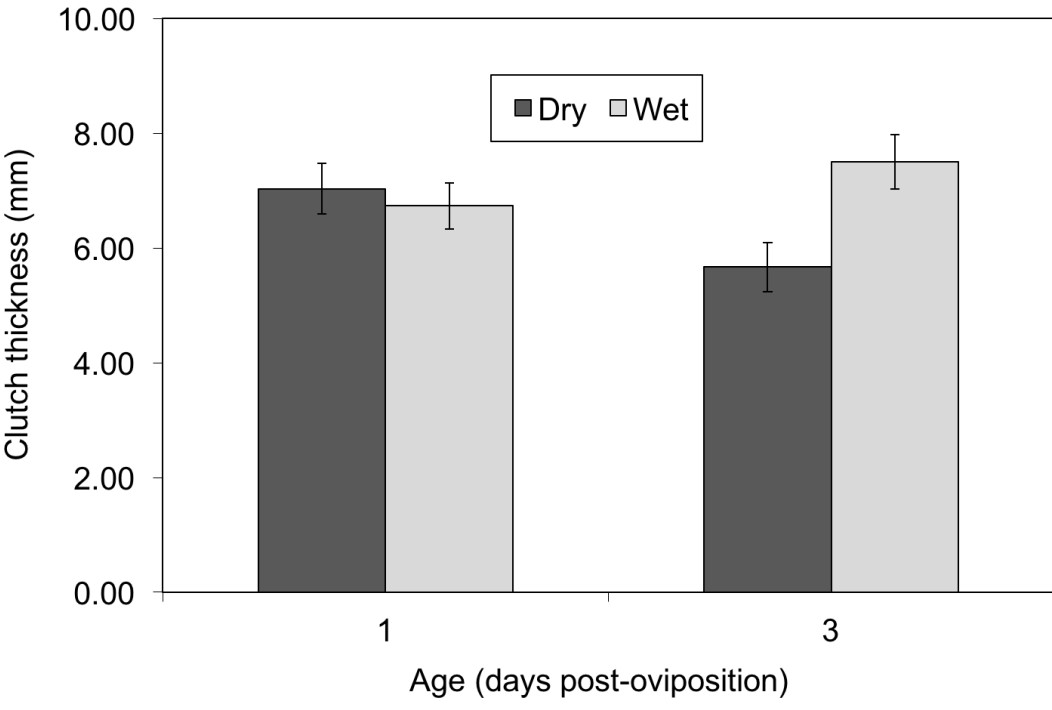

**Figure 5** Thickness of *Agalychnis callidryas* egg clutches before (age 1 d) and two days after (age 3 d) the imposition of different hydration treatments. Data are means ± SE across 15 clutches per treatment. Mean clutch thickness was a function of the interaction between hydration treatment and age.

diameter then begins to decrease, constricting the perivitelline space. Egg diameter of terrestrial breeding frogs usually decreases when they are exposed to dry conditions (e.g., *Kurixalus eiffinger*, *Kam, Yen & Hsu, 1998*; *Bryobatrachus nimbus*, *Mitchell, 2002*), due to the semipermeable nature of their vitelline membrane (*Salthe, 1965*).

With this study, early hatching responses to escape mortality from egg dehydration are now known from three anuran lineages with independent origins of terrestrial eggs (Hylidae: Dendropsophinae; Centrolenidae; Phyllomedusidae). Other responses to, and effects of, dehydration on terrestrial frog eggs have been explored in other lineages. For example, *Kam, Yen & Hsu (1998)* found the opposite response in *Kurixalus eiffinger* (Rhacophoridae: Rhacophorinae); well-hydrated eggs hatched earlier than drier eggs. In this species, accelerating the time of hatching under wetter conditions has a clear adaptive significance. Tadpoles of *K. eiffinger* are oophagous. Females lay their first batch of trophic eggs before all the fertilized eggs have hatched, then return eight days later to feed the tadpoles again (*Kam et al., 1998*). Tadpoles that hatch earlier obtain more trophic eggs, grow faster and reach metamorphosis earlier. Other studies have been conducted on terrestrial anuran embryos with a similar approach. Most of this research has focused on effects of different moisture conditions on phenotypic traits (*Taigen, Pough & Stewart, 1984*; *Bradford & Seymour, 1988*; *Seymour, Geiser & Bradford, 1991a*; *Seymour, Geiser & Bradford, 1991b*; *Kam, Yen & Hsu, 1998*; *Mitchell, 2002*). Anuran embryos exposed to dry conditions grow more slowly (*Pseudophryne bibroni*, *Bradford & Seymour, 1988*), have lower hatching success (e.g., *Kurixalus eiffingeri*, *Kam, Yen*

& Hsu, 1998; *Bryobatrachus nimbus, Mitchell, 2002*), produce smaller hatchlings (e.g., *Eleutherodactylus coqui, Taigen, Pough & Stewart, 1984*; *Kam, Yen & Hsu, 1998*; *Mitchell, 2002* and generate stunted and asymmetric morphologies at hatching (*Mitchell, 2002*). In *A. callidryas,* early-induced hatchlings are generally smaller and less developed than full term hatchlings (*Warkentin, 1995*; *Warkentin, 1999*; *Gomez-Mestre, Wiens & Warkentin, 2008*). Such differences, however, appear simply to be caused by differences in the period of embryonic development, not by differences in embryonic developmental trajectories, and there is no evidence to date that hatching plasticity in this species occurs by altering the rate of embryo development (*Warkentin, 2011a*). Nonetheless, in this study we did not collect the detailed morphological data that would be necessary to test for subtle effects of drying on development rate.

The anuran lineages now demonstrated to hatch early in response to drying vary in their degree of egg and clutch adaptation to terrestrial development. *Dendropsophus ebraccatus* egg size and clutch morphology are much like those of aquatic breeding congeners; they appear not to be strongly adapted to terrestrial development, and indeed can also develop aquatically (*Touchon & Warkentin, 2008*). In contrast, phyllomedusids have a long (34–50 million years) evolutionary history of terrestrial eggs (*Gomez-Mestre, Pyron & Wiens, 2012*) and *A. callidryas* eggs do not survive prolonged submergence (*Pyburn, 1970*). Considering that these highly adapted terrestrial eggs, which typically do not suffer high dehydration mortality, can show adaptive plastic responses to reduce mortality from this occasional threat, drying-induced early hatching may be a more general, broadly distributed phenomenon.

The risk of dehydration as a source of mortality for terrestrial-breeding frogs is particularly important in the context of global climate change. Local changes in weather and climate can affect the hydration of terrestrial embryos. In *D. ebraccatus* living in sympatry with *A. callidryas*, the survival of terrestrial eggs is affected both directly and indirectly by the amount of rainfall (*Touchon & Warkentin, 2009*). Directly, rain hydrates eggs and prevents mortality from drying. Indirectly, because the jelly surrounding eggs swells with hydration, rain decreases the risk of predation; dehydrated eggs are more susceptible to predation by ants and wasps. In *Phyllomedusa hypochondrialis*, which normally wraps its eggs in leaves, the mortality of embryos exposed directly to the air decreased during rainy periods (*Pyburn, 1980*). The tropics, where the highest biodiversity of amphibians is concentrated, are expected to become warmer and drier, and many tropical anuran lineages have evolved terrestrial eggs. Therefore, to understand how these terrestrial eggs can respond to, or will be affected by, rapid changes in climate over the next decades is relevant for conservation planning.

## ACKNOWLEDGEMENTS

This paper is dedicated to the memory of Monique Halloy, advisor to MJS, with gratitude for her support both personally and in herpetological research. We thank J Charbonnier, R Jimenez, S Abinette, S Bouchard, K Cohen and M Leavy for assistance, and the editor and reviewer for comments and suggestions which improved the manuscript.

### Funding

This work was funded by STRI, the US National Science Foundation (DEB-0716923 and DEB-0717220 to KMW and JRV), Boston University and Virginia Commonwealth University. MJS was supported by a Women-in-Networks grant from BU Women in Science and Engineering. The funders had no role in study design, data collection and analysis, decision to publish, or preparation of the manuscript.

### Grant Disclosures

The following grant information was disclosed by the authors:
STRI.
US National Science Foundation: DEB-0716923, DEB-0717220.
Boston University and Virginia Commonwealth University.
Women-in-Networks.

### Competing Interests

The authors declare there are no competing interests.

### Author Contributions

- María José Salica conceived and designed the experiments, performed the experiments, analyzed the data, wrote the paper, prepared figures and/or tables.
- James R. Vonesh analyzed the data, contributed reagents/materials/analysis tools, reviewed drafts of the paper.
- Karen M. Warkentin conceived and designed the experiments, analyzed the data, contributed reagents/materials/analysis tools, wrote the paper, prepared figures and/or tables.

### Animal Ethics

The following information was supplied relating to ethical approvals (i.e., approving body and any reference numbers):

This research was conducted with IACUC approval 100625-1008-15 from the Smithsonian Tropical Research Institute (STRI).

### Field Study Permissions

The following information was supplied relating to field study approvals (i.e., approving body and any reference numbers):

This research was conducted under permit SC/A-13-11 from the Panama Autoridad Nacional del Ambiente.

### Data Availability

The raw data has been supplied as a Supplementary File.

## Supplemental Information

Supplemental information for this article can be found online at http://dx.doi.org/10.7717/peerj.3549#supplemental-information.

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
