# Peer review of "Egg clutch dehydration induces early hatching in red-eyed treefrogs, Agalychnis callidryas"

_PeerJ, doi:10.7717/peerj.3549_

## Round 0.1 · original submission · Minor Revisions

Both reviewers indicated that they found this study to be well designed and clearly presented. Reviewer 1 had several suggestions and requests for clarification and/or justification of some aspects of the statistical analysis, while Reviewer 2 was satisfied with the manuscript as presented. I strongly agree that the study was generally well done and the writing style excellent. However, I did have a small number of minor grammatical suggestions (noted in an annotated pdf) and some questions concerning the methods and statistical analysis (noted below).

Comments from the editor:

L125. You imply that you know the temperature and humidity in the lab and field, so it would be appropriate to provide some measure of the mean and range of these values.
L134ff. It appears to me that your design used clutches rather than eggs as the experimental units. Please consider whether not taking clutch into account is appropriate for calculating the statistical probability and effect size for hatching time, egg size and mortality.
L143. How did you estimate 11% difference? Can you provide supporting values such as the number of days to median hatch time for each treatment?
L149 and elsewhere. I believe that SD rather than SE is usually recommended for describing variation of descriptive data.
L365. It would be helpful to have the total number of eggs in each treatment mentioned in the caption. If clutches are the units, this figure might change to something like the mean of proportion hatched per clutch.

·

Basic reporting

No comments

Experimental design

No comments

Validity of the findings

No comments

Additional comments

Salica et al.’s paper, “Egg clutch dehydration induces early hatching in red-eyed treefrogs, Agalychnis callidryas”, is a very nice study of the effect of moisture on time to hatching in a treefrog species. The authors hypothesize, based on work in other species, that embryos experiencing dehydration will hatch earlier than embryos that remain well-hydrated. The study is a simple factorial design in which eggs are hydrated regularly or allowed to dry, and the embryos are followed over time until they hatch. The authors found that embryos allowed to dehydrate hatched sooner.

I find this to be a very nice and simple study. I have a few questions about the implications and explanation. I am curious to know whether the stage at hatching was the same for both treatments. In the discussion, the authors say that there is no evidence that there is a difference in developmental rate between treatments, but I don’t think they have data to support this argument from this study. The relevance is that there is considerable information about the importance of hypoxia for triggering hatching in amphibians (Petranka et al. 1982; Warkentin papers) and the importance of oxygen in developmental rate. Together, there are conflicts: hypoxia causes slower development but it also triggers hatching, possibly resulting in a complicated prediction for how oxygen should effect time to hatching (examined indirectly in Hale et al. 2016). So, is the difference in time to hatching mediated by some effect of hydration on oxygen availability? Perhaps the dehydrated embryos hatch because they reach a developmental stage that requires more oxygen than they can get through their drying membranes, whereas hydrated embyros continue to get sufficient oxygen for development through to a later stage. I acknowledge that this study was not designed to address the mechanisms of the difference in development time, but I think a discussion of these issues would be nice, given the expertise of the coauthors.

Specific comments:
Line 85 – Unnecessary capitalization of “Earlier And More Synchronously”
Line 87 - I'm a bit confused about whether terrestrial and dry conditions are to be considered synonymous. A few species of Ambystoma (A. opacum and A. cingulatum) have terrestrial eggs (e.g., Petranka 1998, Hale et al. 2016), but I don't think I'd consider those nests to be dry. So, are these species breeding in conditions in a terrestrial environment that are especially dry, or that dry out over time?
Line 114 - Please give information about the average size of clutches. How many eggs are you talking about?
Line 119 – Why pair the clutches? Pair ID is never addressed later, or used in the analyses as a random effect.
Line 134 – Version 2.3.1?
Line 136 - Time to hatching isn't binary, so why was a binomial error indicated? In general, I’m confused by this binomial GLM. Is this a logistic regression? I don’t see how logistic regression the appropriate test here since time would be the independent variable and the data aren’t independent from one time step to the next. Proportion hatched is not the appropriate dependent variable for logistic regression, so I think I’m misunderstanding. Please clarify.
Line 137 - Why 2-way anova? Is pair ID considered a block effect? If so, why here and not for egg mortality? It’s apparent in the Results that the second variable is ‘day’. Please state that here and explain whether the binomial regression is also 2-way.
Line 159 – Replace the comma with a colon.
Line 196 – In my experience, it is not appropriate to re-cite a figure in the discussion.

·

Basic reporting

The manuscript is clearly written with detailed attention to the published literature and context.

Experimental design

The experimental design is described fully and appropriate controls are in place. The animals were treated humanely. The study could be replicated by others with the information in the manuscript.

Validity of the findings

The results are presented with sufficient support. The graphs are professional quality. The statistics are appropriate and presented with the raw data. The authors address potential confounding variables such as the rate of embryonic development and the thickness and water content of the gel layer surrounding the eggs. The phylogenetic significance is discussed, and is a valid reason for the exploration of this species as compared to previous research on this phenomenon.

Additional comments

I enjoyed reading this interesting manuscript. I have no recommendations for improvement. You answered my questions in your text. Well done!

---

## Round 0.2 · accepted · Accept

The authors have done a very good job of addressing the issues raised by the reviewer and by me. The manuscript is now ready for publication.